# Position: Mechanisms for *Aggregated Individual Reporting* Should be Established for Post-Deployment Evaluation

**Jessica Dai** [1]  **Inioluwa Deborah Raji** [1]  **Benjamin Recht** [1]  **Irene Y. Chen** [1 2]

## Abstract

The need for developing model evaluations beyond static benchmarks, especially in the post-deployment phase, is now well-understood. At the same time, concerns about the concentration of power due to AI systems have sparked a keen interest in "democratic" or "public" AI. In this work, we bring these two ideas together by proposing mechanisms for *aggregated individual reporting* (AIR), a framework for post-deployment evaluation that relies on individual reports from the public. An AIR mechanism allows those who interact with a deployed (AI) system to report when they feel that they may have experienced something problematic; these reports are aggregated over time, with the goal of evaluating the relevant system in a fine-grained manner. This position paper argues that **individual experiences should be understood as an integral part of post-deployment evaluation, and that the scope of our proposed aggregated individual reporting mechanism is a practical path to that end.** On the one hand, individual reporting identifies substantively novel insights about safety and performance; on the other, aggregation is uniquely useful for informing action. From a normative perspective, the post-deployment phase completes a missing piece in the conversation about "democratic" AI. As a pathway to implementation, we provide a workflow of concrete design decisions and pointers to areas for future research.

## 1. Introduction

In the third week of April 2025, OpenAI quietly pushed an update to GPT-4o, the model that powers the ChatGPT product by default. Online, the complaints started rolling in: The newest update was annoying and introduced bugs; it overpromised and underdelivered. More frighteningly, it encouraged users to stop taking their medications, validated conspiracy theories, and worse. By April 29, around one week later, OpenAI had rolled back the change[1].

In this case, the unstructured feedback of individual users was essential. OpenAI was unable to identify ahead of time that this "personality" update was problematic—in part because it is hard to anticipate the richness of usage patterns, and therefore failure modes. Users thought the problems were egregious enough that it motivated them to share online; enough users tweeted about the *same* problem that OpenAI noticed. This ChatGPT personality problem was serious, widespread, and was therefore caught even in the absence of a formal system to collect feedback. But what other, subtler, non-Twitter-viral patterns might be happening—and how might we find out?

In this work, we formalize *aggregated individual reporting* (AIR) as a general framework for understanding the real-world impact of an AI system in a structured, intentional, and thorough manner. At a high level, an AIR allows those who interact with a specific AI system to provide feedback (reports) when they believe they have experienced harm due to the system. The mechanism aggregates these reports over time, with the goal of building collective knowledge about the contours of system behavior. One person having one bad experience does not by itself necessarily imply a system-level problem. On the other hand, if many reports of similar experiences begin to accumulate, then perhaps there may be an important underlying issue that requires action.

Our framework relies on two crucial beliefs: first, that those interacting with AI systems have unique and valuable perspectives on its failure modes, and second, that they ought to be able to express those perspectives in a nontrivial way. **This position paper argues that post-deployment evaluation of AI systems must account for individual experiences—and that *aggregated individual reporting mechanisms*, as we define in this work, are a practical pathway to doing so.**

[1]University of California, Berkeley [2]University of California, San Francisco. Correspondence to: Jessica Dai <jessicadai@berkeley.edu>.

*Proceedings of the 43rd International Conference on Machine Learning*, Seoul, South Korea. PMLR 306, 2026. Copyright 2026 by the author(s).

---

[1]openai.com/index/sycophancy-in-gpt-4o/

We do not claim to originate the concept of individual reporting. In fact, we are inspired by the existence of similar reporting mechanisms in other domains, like the Vaccine Adverse Events Reporting System (VAERS) (Shimabukuro et al., 2015), the Aviation Safety Reporting System (ASRS) (Beaubien & Baker, 2002), and various medical reporting systems (Wu et al., 2002). We are also heavily influenced by calls to incorporate end-user expertise in algorithm evaluations (e.g., Shen et al. (2021); DeVos et al. (2022); Lam et al. (2022)) and to shift power towards the public (e.g., Kalluri (2020); Feffer et al. (2023)). Moreover, while several recent policy directives call for considering public feedback with specific reference to post-deployment monitoring (e.g., the EU AI Act (European Parliament, 2024), the UN General Assembly's first AI resolution (UN General Assembly, 2024), and Biden's now-repealed executive order (Biden, 2023)), they are, by nature, only high level. Our work aims to offer a shared language as a starting point for conversation across subfields and domains, and to concretize what effective implementation of these policies might look like.

We also highlight two recent concurrent works that advocate similar proposals, but which offer complementary substantive contributions. Longpre et al. (2025) gives a thorough legal and policy overview and a taxonomy of current flaw disclosure methods, which our manuscript excludes, as well as a sample reporting workflow. Meanwhile, Gailmard et al. (2025) conducts a thorough analysis of the reporting systems in other domains (e.g., public health and transportation) that we mention only briefly, and discusses design principles in light of them; they also provide more extensive perspectives on the organizational challenges in designing regulation and industry norms. Our work covers aspects that are not the focus of these works, and we believe it is worthwhile to consider all three perspectives in parallel.

The remainder of this paper is organized as follows. In Section 2, we define our idealized AIR mechanism, and contextualize our definition with the current state-of-the-art for post-deployment evaluation and auditing for AI systems. In Section 3, we argue that individual reporting effectively identifies "unknown unknowns," and that aggregation enables downstream action. In Section 4, we elaborate on the normative case for individual reporting, placing our proposal in the context of recent calls for "democratic" AI. Section 5 describes more granular design decisions and associated open research questions. We conclude by discussing challenges for successful deployment in Section 6.

## 2. Defining aggregated individual reporting

Our idealized vision of a mechanism for aggregated individual reporting, illustrated in Figure 1, is intentionally broad. Even with this broad scope, we note that, to the best of our knowledge, (public or non-proprietary) AIRs for AI systems are not currently in mainstream use.

### 2.1. Defining an idealized AIR

An AIR mechanism comprises three key entities. First, the **evaluated system** determines the scope of the AIR mechanism. For example, this could be a general-purpose model like ChatGPT or Claude; a product that scribes in-person doctors' appointments; or a predictive algorithm that several banks use for making loan decisions. Reports are submitted *about* the evaluated system. Second, the evaluated system induces an **affected population**. This could include users of ChatGPT, or their loved ones; patients and healthcare workers; or loan applicants. Reports are submitted *by* members of the affected population. Finally, the AIR is orchestrated by a **mechanism administrator**, which is the organizational entity that collects and aggregates the reports. The administrator makes key decisions like determining the scope of the evaluated system and the affected population, as well as various implementation details as discussed in Section 5.

These entities interact through three core components that define an AIR mechanism:

(1) *Individual reporting*: Reports are submitted by members of the affected population about specific experiences with the evaluated system.

(2) *Aggregation for evaluation*: Reports are aggregated and interpreted over time. The goal of such aggregation is evaluation: to understand or describe the behavior of the evaluated system in a fine-grained manner.

(3) *Evaluation-conditional action:* The aggregated evaluation supports downstream action. That is, there are evaluation outcomes where, if and when the reports are consistent with those outcomes, the mechanism administrator takes associated action.

Ensuring that reports are tied to the *evaluated system* makes it possible for aggregation to generate specific insights about system behavior; this specificity allows for downstream action, in that it explicitly describes some (undesirable) property of the system. The temporal component of continuous evaluations makes it possible to understand changing use cases and experiences over time, and moreover, to identify problems (and take action) quickly as they arise.

In Figure 2, we give four examples of AIR mechanisms. In the first row, we show how VAERS, an existing reporting system, implements the criteria of our framework. The second row presents a hypothetical problem setting following an example in Dai et al. (2025a); the third describes a real-world application studied in Dai et al. (2025b); and the fourth describes a speculative example of an AIR application. As these examples indicate, AIRs can be established for various types of evaluated systems; moreover, the implementation of the mechanism depends closely on the type of

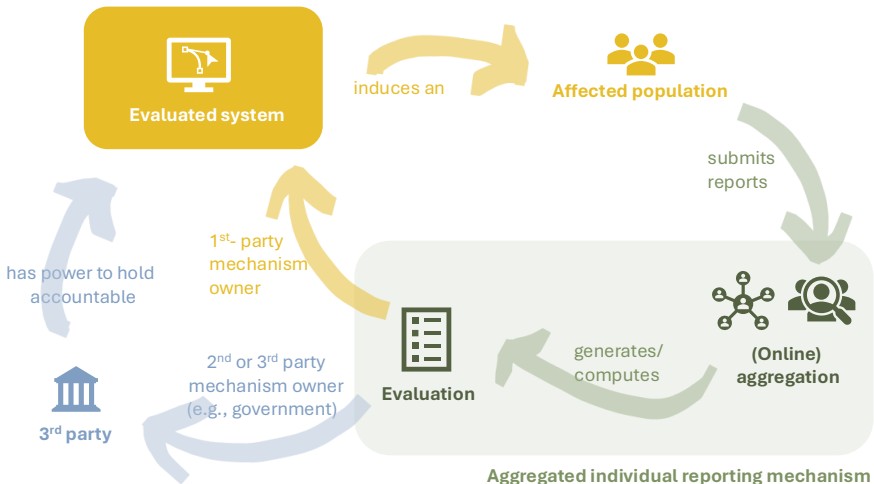

*Figure 1.* The components of our framework and how they interact. Any *evaluated system* will have an associated *affected population*. Members of this population can submit reports to the AIR, which uses collected reports to compute an evaluation that reflects the aggregated experiences among submissions. Potential downstream actions may depend on who the *mechanism administrator* is.

harm that an AIR is designed to surface.

## 2.2. Current state-of-the-art in post-deployment evals

The idealized definition given in Section 2.1 excludes most prominent systems that currently exist for crowdsourcing and post-deployment monitoring for AI. Existing approaches generally fall into three categories: third-party collections of real-world problems with AI systems; targeted flaw disclosure approaches like bug bounties and red-teaming; and more general approaches to post-deployment evaluation. We briefly clarify the relationship of AIRs to the first two, and discuss the third in Appendix A.

In the first category are several well-known *incident databases and risk repositories*, which include the website `incidentdatabase.ai`, where incident submissions are available to the public (McGregor, 2021); the OECD's AI Incidents and hazards Monitor (AIM), which scrapes AI-related news headlines globally (OECD, 2023); and the MIT AI Risk Repository, which tags individual incidents from `incidentdatabase.ai` with additional metadata from the Risk Repository framework (Slattery et al., 2026) and is available online as a Google sheet. While the exact implementation of each database is slightly different, the high-level commonality is that they tend to collect discrete, externally-submitted *incidents*. These incidents are typically news events that were related in some way to *any* deployed AI system, with a general focus on examples of misuse.[2] As

a result, these collections of incidents are much broader than individual interactions that can become descriptive of model behavior; in fact, aggregation of these incidents is possible only to the extent of tabulating approximate frequencies of discrete news events across impact category and severity.

The second category includes *flaw disclosure* mechanisms, e.g., as outlined in Longpre et al. (2025); these are an important starting point, but are insufficient for our goals. That is, an aggregated individual reporting mechanism can—and likely should—include much of the flaw disclosure machinery outlined in Longpre et al. (2025), but a flaw disclosure system by itself would not necessarily qualify as an AIR. In the context of our definition, flaw disclosure systems may sometimes satisfy the first condition (*individual reporting*), but do not include the other criteria (aggregation for evaluation and downstream action). Thus, our specific proposal is explicitly narrower. This can be seen when considering concrete instantiations of flaw disclosure mechanisms, such as bug bounties for algorithmic problems and red-teaming. The former is typically focused on resolving bugs on a per-report level, rather than understanding problems and taking action on an aggregate level (e.g., as discussed in Kenway et al. (2022), with the bias-bounty mechanism of Globus-Harris et al. (2022) as a rare exception); the latter often focuses on the pre-deployment phase, and solicits participation from predefined groups rather than all members of the affected population (e.g., Ahmad et al. (2025)).

Our proposal is complementary to all of these approaches, which are useful and important parts of the AI evaluation

---

[2]For instance, at time of writing, the most recent incident at `incidentdatabase.ai` is that New Orleans police used banned facial-recognition software (Incident 1075). The harm being described in this incident is about NOPD's usage of a particular

system, rather than a specific instance of that system's failure.

| Evaluated system | Mechanism administrator | Affected population | Individual report information | Evaluation condition (when would downstream action be taken due to patterns in reports?) | Downstream actions |
|---|---|---|---|---|---|
| **FDA-approved vaccines** (deployed in real-world system as VAERS) | 3rd-party (United States CDC & FDA) | All patients who received a particular vaccine | Vaccine info, specific adverse event, and demographic information | Elevated overall frequency of adverse event reports compared to baseline frequency *e.g., myocarditis appears frequently for the COVID-19 vaccine.* | Further investigation of specific vaccine-side effect pairs (e.g. additional research or data collection), and notification of relevant parties (e.g., press release) |
| **Loan allocation algorithm at Bank X** (hypothetical from Dai et al. 2025a) | 3rd-party (nonprofit organization) | All loan applicants to Bank X | Demographic information and claim of potential discrimination | Identification of a subgroup that experiences disproportionate rates of harm *e.g., financially-healthy Black applicants are denied loans at a higher rate.* | Gathering evidence to (e.g.) initiate a legal discrimination case |
| **AI medical scribe product** (studied in Dai et al. 2025b) | 2nd-party (hospital system client) | Healthcare workers who use the tool | Free-text notes; scribe text and edit history | Clinically-relevant failure modes of scribe product. *e.g., AI scribe tends to make errors when medications must be taken on a schedule.* | Feedback provided to company developing AI scribe; usage guidelines given to clinicians |
| **ChatGPT** (speculative example) | 1st-party (OpenAI) | All users of the ChatGPT product | Chat transcript and free-text notes | Wide-scale safety-critical behavior. *e.g., the newest model exhibits dangerously sycophantic behavior.* | Rollback to prior model version; post-mortem conducted for flaws in internal evaluations. |

*Figure 2. Examples of how AIRs could be set up for a variety of applications. We elide the corresponding aggregation methods in order to focus on the application itself; note that the "evaluation conditions" are aggregate system failures rather than per-report problems.*

ecosystem. AIRs, as we have defined them, enable distinct types of evaluations beyond what is already covered by these efforts—but cannot, by themselves, supersede them.

## 3. Aggregated individual reporting enables actionable harm discovery

In this section, we argue that AIRs have concrete benefits beyond what existing systems can provide. Prior work, especially in human-computer interaction, has documented a desire for ways to enable auditing and evaluation of algorithmic systems from a wider population. For example, user-driven auditing (e.g., Shen et al. (2021); DeVos et al. (2022); Deng et al. (2023); Lam et al. (2022)) has been studied as a means for eliciting user feedback that identifies problems with an algorithmic system beyond centralized evaluations. Shared concerns and challenges presented by these works, which have primarily involved small-scale case studies, include aggregating and interpreting feedback, and using feedback to drive downstream action. In fact, Ojewale et al. (2025) explicitly call for the development of "tools for harm discovery" and "participatory methods" as a pathway towards accountability.

Our framework seeks to formalize these intuitions; each component of our definition plays a specific role. Individual reports enable harm discovery, while aggregation is a prerequisite to making them actionable.

### 3.1. Individual reports surface unknown unknowns

In domains where reporting systems are well-established, it is widely understood that reports contain useful information that may otherwise never be identified (e.g., see Beaubien & Baker (2002) for a decades-old review in aviation, and Wu

et al. (2002) for health). In our vision of individual reporting, however, the affected population is a much wider set of people, beyond domain experts and practitioners. What kinds of insights might we expect to see from their reports?

Prior work suggests that audit-style feedback from "end users" draws from their prior beliefs and experiences, and can reveal clusters of shared problems (including some that were previously unknown), as well as identify meaningful disagreement across users (Lam et al., 2022; DeVos et al., 2022). For concreteness, we focus here on social media platforms like Twitter/X as case studies for (informal) individual reporting. Twitter is of course not a purpose-built reporting system, and there is no aggregation mechanism dedicated to explicitly making sense of report content. However, individuals frequently take to social media to share their experiences with specific algorithms. Shen et al. (2021) term this phenomenon "everyday algorithm auditing," and show how social media platforms enable an organic, informal process for both raising awareness and validating problems raised by initial reports. In their analysis, they highlight that "everyday audits" leverage the lived experience (e.g., cultural background) and situated knowledge (e.g., application contexts) of everyday users.

Returning to the motivating example at the beginning of this paper, the analysis of Shen et al. (2021) is largely consistent with the patterns that can be seen from posts about ChatGPT sycophancy. For example, users noted performance degradation even in quantitative tasks[3], reflecting situated knowledge with respect to users' expectations in specific domain applications; more serious posts highlighted scenarios

---

[3] x.com/Eddie_Ho2025/status/1913596114156028013 and x.com/IndieLauraSDG/status/1913102627837206587

where ChatGPT validated flat-earth and conspiracy beliefs[4], reflecting both lived experience and situated knowledge.

A subtly distinct pattern in these sycophancy-related reports, compared to trends identified by Shen et al. (2021), is reports that reflect user creativity—rather than realistic practical usage—in a way that can be thought of as informal "red teaming." For example, users showed example responses when given prompts about math and "pickle rick", monologues by unsavory fictional characters, as well as genuinely safety-critical responses elicited intentionally[5].

Importantly, the sycophancy case study also reflects a limitation of relying on social media as a vehicle for "reporting." The *types* of problems that can be identified via these "reports," somewhat by definition, are those that are amenable to virality. In fact, the methodology of the taxonomy in Shen et al. (2021) also relied on prior knowledge of cases that were "high-profile" on social media. Sycophancy is a prime example of a virality-friendly problem: It appeared to affect all users across use cases and backgrounds, so that various users felt empowered to share their particular experience that reflected the underlying problem. Moreover, it was easy to produce content that illustrated sycophancy, but was also (e.g.) highly humorous or inflammatory. However, many serious model flaws are not so "clickworthy"; some failures might only affect a small slice of users, and in ways that cannot be constructed as a popular tweet.

Even when analysis is not limited to posts primarily about "auditing", social media appears to contain useful information about real-world usage; for instance, Dai et al. (2026) analyze posts on r/ChatGPT and find that posts describing usage related to therapy, companionship, or emotional support appear to grow substantially after the release of GPT-4o in May 2024—and that this pattern would have counterfactually been discoverable in near real-time. However, stronger claims about causality or absolute prevalence are impossible, especially in the absence of "ground-truth" usage data. More systematic approaches, therefore, are necessary.

### 3.2. Aggregation for actionability and accountability

We now argue that, beyond individual reporting, *aggregation* is necessary to make use of the information contained in reports. Not all potential problems with an evaluated system are inherently statistical; in many cases, however, individual experiences can only be understood in the context of aggregated evaluations. For example, if the goal is to identify subgroups that disproportionately experience harm, as in Dai et al. (2025a), then individual reports become significant insofar as they can be described statistically; similarly, participants in DeVos et al. (2022)'s study mention uncertainty about whether particular algorithmic behaviors ought to be considered examples of harm, especially without knowledge about how others experienced the system.

For areas where report databases (often called "incident databases") are already established as a component of safety monitoring, it is well-understood that focusing on rectifying individual incidents is limited. Instead, the goal should be to find high-level patterns that can inform future procedural changes; in these fields, learning from reports has indeed been effective for shaping future practice (Macrae, 2016; Jacobsson et al., 2012; Robinson, 2019).

Despite this, many reporting systems do not include intentional aggregation as a core component; these systems appear to have had limited impact beyond the scope of reports themselves. On the other hand, the few examples of successful action from crowdsourced information have involved aggregation, suggesting that aggregation is a necessary (though potentially not sufficient) condition for taking high-level action from reports. In the remainder of this section, we discuss several case studies of crowdsourced or public reporting feedback, and the extent to which they successfully spurred concrete action.

**CFPB Consumer Complaints, VAERS, and different levels of actionability.** The U.S. Consumer Finance Protection Bureau (CFPB) maintains a Consumer Complaint Database to which members of the public are eligible to submit[6]. The CFPB itself does not address these complaints; instead, it acts as an intermediary, passing the complaints to the relevant financial institutions, which are mandated to respond (Littwin, 2015). Though the complaints submitted by individuals do actually receive responses from the responsible parties (i.e. banks), the emphasis is on directly addressing individual incidents, rather than the problems that emerge when considering them all collectively.

To the best of our knowledge, insights from the CFPB complaint database have not triggered further enforcement or legislative action; this is perhaps unsurprising, given the emphasis on actionability for individual complaints, rather than in aggregate. Several academic works have found problematic trends by analyzing complaints collectively (e.g., Bastani et al. (2019); Ayres et al. (2013); Haendler & Heimer (2021)); thus, this focus on per-complaint resolution is a design choice, not an inherent limitation of the data itself.

On the other hand, the VAERS database is continually mon-

---

[4] www.reddit.com/r/OpenAI/comments/
1kasjmr/this_new_update_is_unacceptable_
and_absolutely/

[5] x.com/williawa/status/
1916935712550551743, x.com/Siddharth87/
status/1916999455146185022, x.com/
ReviewsPossum/status/1917292836082397355,
x.com/___frye/status/1916346474893656572

[6] www.consumerfinance.gov/data-research/
consumer-complaints/

itored explicitly *for* aggregate-level harm. For VAERS, the event of concern is not any individual report of an adverse event. Rather, the concern is whether reports for particular vaccines occur with abnormally high frequency; examples of clinically-relevant side effects initially discovered via VAERS abound (Shimabukuro et al., 2015; Singleton et al., 1999). Beyond initial identification, another important usage of VAERS has been post-hoc investigation of problems that were originally flagged via case study. For instance, it is now well-known that the COVID-19 vaccines induced an elevated risk of myocarditis, but only in younger men; however, in the early stages of vaccine rollout, the conversation was limited to various healthcare providers noticing, via case study, that myocarditis appeared to be a common occurrence overall (Mouch et al., 2021; Larson et al., 2021; Marshall et al., 2021). A more holistic understanding of the issue, including that myocarditis appeared to be limited to younger men, was attained only after post-hoc analysis of VAERS reports (Witberg et al., 2021; Oster et al., 2022). This, in fact, was one motivating example for the method proposed in Dai et al. (2025a): if VAERS reports about myocarditis had been analyzed with a disaggregation over demographic identity, then VAERS reports could have directly confirmed the affected subgroup more quickly.

**ChatGPT sycophancy and the limitations of informal aggregation.** We conjecture that one reason that the Chat-GPT sycophancy problem resulted in decisive change was that the Twitter timeline algorithm, and the platform's affordances of likes and retweets, served as a quasi-aggregation scheme. The brief dominance of tweets highlighting sycophancy on the timeline showed that this was a problem that impacted a wide range of users, and that there was general agreement that the behavior was problematic. In a move that is remarkably unique for tech companies, this culminated in explicit action by OpenAI to roll back the new model deployment, and to initiate some discussion of what went wrong[7]. In some sense, it was "lucky" that this particular safety problem happened to have been friendly for virality; in general, there is no guarantee that quasi-reports to social media are necessarily seen, or taken seriously, by those who have control over the evaluated system.

**Targeted crowdsourcing and the value of statistical evidence.** Finally, we discuss two systems that crowdsourced data for evaluating specific algorithmic systems as positive examples of aggregation. While the scope of these surveys was narrower—meant to discover average rates of pre-specified metrics, rather than general evaluation—they are notable because their aggregations provided concrete statistical evidence for individual experiences.

In 2020 and 2021, Mozilla led a study using a browser extension called *RegretsReporter*, which crowdsourced information about the YouTube recommendation algorithm (McCrosky & Geurkink, 2021). The study found that recommendations from the YouTube algorithm were disproportionately responsible for serving content that users regretted seeing (and that violated terms of service)—and moreover, that non-English speakers were most seriously affected.[8] While YouTube never disclosed whether specific algorithmic changes were made (a weakness we discuss in Section 6), the company did address the report in public statements[9]. Meanwhile, *FairFare* is a system that crowdsources information on rideshare wages, with the goal of understanding the extent to which drivers were being underpaid as well as overall average pay rates (Calacci et al., 2026). The statistics computed via FairFare not only empowered drivers to organize, but also led to direct legislative impact.

More broadly, statistical evidence—in contrast to, e.g., anecdotal accounts—is especially useful as a means for pressuring *organizations* or *institutions* to make change (Recht, 2025). For example, business leaders are more amenable to making decisions that appear "data-driven," rather than responsive to anecdotal experiences (Davenport, 2014). Statistical evidence is also treated differently in litigation contexts (see, e.g., Espeland & Vannebo (2007)). The question of what *kinds* of statistical evidence are considered acceptable will depend on the particular application context—and, of course, not all harms are inherently statistical. At the very least, however, the language of statistics can empower individuals by validating their experiences. Both RegretsReporter and FairFare were simply *formalizing* folk intuitions that YouTube users and rideshare drivers individually already had—but the collection and aggregation of data made it impossible to dismiss those perspectives as individual experiences of one-off anomalies.

## 4. Aggregated individual reporting as a pathway towards "democratic" AI

In this section, we briefly discuss the normative underpinnings of our proposal. In recent years, the recognition that modern AI systems both require and accelerate the concentration of power has spurred a flurry of research on how AI systems might be designed through quasi-democratic processes—"participatory" (Birhane et al., 2022; Delgado et al., 2023; Gilman, 2023), "pluralistic" (Gölz et al., 2025;

---

[7]openai.com/index/sycophancy-in-gpt-4o/, openai.com/index/expanding-on-sycophancy/

[8]The latter is also an example of crowdsourced data finding new, non-obvious insights, as discussed in Section 3.1.

[9]thehill.com/policy/technology/561788-youtube-algorithm-keeps-recommending-regrettable-videos/, www.theverge.com/2021/7/7/22567640/youtube-algorithm-suggestions-radicalization-mozilla,

Dai & Fleisig, 2024; Sorensen et al., 2024a;b), "collective" (Huang et al., 2024), and so on. Methodological research in these directions focuses almost exclusively on the development phase, and rarely considers how "the public" might engage with AI systems post-deployment—despite calls for increasing the power of "the public" from more conceptual work (e.g., Feffer et al. (2023)), and for explicitly "democratic" governance processes (e.g., Ovadya et al. (2025); see Appendix A for additional discussion of the Democracy Levels framework in relation to our proposal.)

The "democracy" analogy relies on a rough analogy between AI systems and governmental bodies (about which democracy is classically theorized). For instance, these works commonly ask, how might "democratic" inputs shape the model spec or training objectives? Implicit in this question is an analogy to the same way that "democratic" inputs determine the outcome of an election. We argue that such an analogy, while not incorrect, is incomplete. A key tenet of democracy is *consent of the governed*: the ability for members of the public to express their will about governance, and a process for those expressions of will to have meaningful bearing on future outcomes.[10] Crucially, such consent is not a singular event; instead, it is an ongoing process that, theoretically, must continue for as long as the governing body remains in power (Bertram, 2023; Gourevitch & Rousseau, 2018). Democratic legitimacy derives not just from the fact that citizens can shape the parameters of their government's future actions, but also from their ability to provide input on ongoing action—to revoke consent.

A critical ingredient for a "democratic" approach to AI, therefore, is the ability for members of the public to collectively raise issues with systems *after* they have already been deployed. In the same way that a democratic governing body (a kratos) *requires* input from its citizens (demos) to continue effective governance, those who operate an AI system cannot fully understand the real-world behavior of their system without the input of those who have interacted with it. And, in the same way that the demos *ought* to have its concerns heard by its kratos, those who interact with possibly-consequential systems also *deserve* for their concerns to be taken seriously by those who build the systems.

In other words, "democratic AI" is not just a design problem; it is an accountability problem. Our framework seeks to take a step towards this by not only offering the ability for individuals to provide feedback—via reporting—but also providing an avenue for them to be heard. As a starting point, aggregation is a lens through which (e.g.) the owners of the evaluated system can understand and interpret feedback ("helping the state see," in the sense of Scott (1998), by surfacing details that centralized evaluations are unable to understand). More theoretically, aggregated reports also

---

[10]This is, e.g., canonically expressed in Locke (1988).

have the potential to develop and shape the "general will," in the sense of Rousseau, from a collection of expressions of "individual wills" (Gourevitch & Rousseau, 2018).

## 5. Towards implementation

We now discuss design decisions for the implementation of any AIR. While these decisions are interdependent—reporting affordances also affect what aggregation methods would be effective, as well as what types of evaluations are available—there are a wide range of ways in which these decisions could be made, depending on context. Each also naturally gives rise to various multidisciplinary research questions, which we discuss in turn; more detailed examples of design decisions are provided in Appendix B (Figures 3, 4). In this section, we use two running examples: FAIRNESS (Dai et al., 2025a), which proposes individual reporting for post-deployment fairness auditing, and provides algorithms that are specific to this task, and SCRIBE (Dai et al., 2025b), which studies feedback from clinician users for evaluating a real-world AI medical scribe product.

### 5.1. Concrete design decisions

**Organizational decisions.** The first core set of decisions that must be made are organizational: what entity will take the role of *mechanism administrator*, and what relationship will it have to the *evaluated system*? This decision affects what kinds of problems the mechanism hopes to identify, as well as the nature of available downstream action.

A first-party administrator will have the fullest information about the evaluated system (e.g., when particular feature rollouts or updates were made), and be able to quickly gather additional data that may become relevant in order to contextualize information raised by reports. Since the first-party administrator has control over the evaluated system, the organization can also directly make changes in response to evaluation results. However, a first-party administrator may also inadvertently restrict the scope of the system, or intentionally choose to ignore the evaluation.

On the other end of the spectrum, a third-party administrator would have nearly no additional information beyond what is included among reports, and cannot directly update or improve the system. However, such an organization could bring external leverage to the evaluated system, e.g. via media or legal pressure. FAIRNESS is developed under an assumption that the goal is to identify the harmed subgroups, but, as a primarily methodological work, it does not specify what organizational entity would be the administrator, or what downstream action might look like; on the other hand, SCRIBE is explicitly set up by a second-party (the hospital system), which does not own the product, but is nevertheless able to directly influence its usage and development.

**Reporting affordances.** What information is included in a report, and how do reporters experience the process of reporting? Examples of potential report formats can be seen in the fourth column of Figure 2, as well as the flaw disclosure worksheet in Longpre et al. (2025). For the algorithm proposed by FAIRNESS, the only information collected in a report is demographic information about the reporter; however, one could naturally imagine that reports could include more information depending on the application, such as medical history (for a vaccine or pharmaceutical system) or financial background information (for a loan allocation system). For example, the system analyzed in SCRIBE contains free-text information from physicians describing mistakes.

**Reporting behavior.** Due to the nature of reporting data it is, intrinsically, essential to understand reporting behavior: what factors affect the decision to submit a report, and, crucially, in what ways might reports be correlated to the target of evaluation? Do different subpopulations report at different rates? Do different types of issues lead to different reporting behaviors? In FAIRNESS, the choice is to commit to a set of (quantitative) assumptions about the extent to which reporting rates can vary, and incorporate those assumptions into the design of the algorithm; in SCRIBE, heterogeneity in clinician feedback behavior is observed empirically (though not yet handled explicitly).

**Aggregation method and evaluation condition.** Given the affordances offered to reporters, what method will be used to interpret them over time—that is, how are evaluation results computed? Given that method, what specific results would trigger downstream action? Ideally, methods should explicitly consider a temporal component. This is because AIRs should be accessible to reporters at any time (rather than within a centralized study with defined start and end dates), as well as the classic distribution shift problem: users' needs in relation to an evaluated system will change over time, as will the system itself.

In FAIRNESS, the method is formalized as a sequential hypothesis test with type-I error control; thus, the evaluation result that triggers action is exactly when the "null hypothesis" of no unfairness can be rejected. A rejection might, e.g., prompt further inquiry into the reported harm for a subgroup. The initial approach in SCRIBE applies clustering in-hindsight, without a sequential component, on recent clinician feedback; the preliminary finding (and evaluation condition) is about the presence of certain types of errors that may affect patient safety.

## 5.2. Active research communities

We highlight some examples of existing lines of work within the AI research community—beyond auditing and evaluation—that can effectively inform these design questions. While these examples are non-exhaustive, we hope to illustrate that AIRs can benefit from a wide range of methodological and disciplinary perspectives.

**Interaction design.** From prior work, we know that the affordances that are available to users to share feedback affect the content they share (e.g., social media platforms and virality-friendly content; see also discussion in DeVos et al. (2022) about the importance of platform affordances). What kinds of designs can encourage the most effective reporting feedback (e.g., can facilitate long term engagement, as discussed in Deng et al. (2023))? Are checklists, as formalized in Longpre et al. (2025), sufficient? Should intermediate evaluation results be made public, and if so, how? E.g., if individuals see that others have reported similar problems, does that encourage them to submit their own reports?

**Reporting rates, incentives, and behavior.** Empirical studies of existing reporting systems have shown that different subgroups often report different types of problems at different rates (e.g., Agostini et al. (2024); Liu & Garg (2022); Liu et al. (2024)); at the same time, these works also suggest the potential for statistical methods that can estimate or correct for varying reporting rates. As the crowdsourcing experiments of Globus-Harris et al. (2024) suggest, participants can be adversarial, and the details of mechanism implementation should be careful to incentivize "good" reporting behavior. Prior work in motivations and incentives of users on online platforms also suggests avenues for study, both qualitatively (e.g., Malinen (2015)) and via theoretical models (e.g., Ghosh & McAfee (2011); Hu et al. (2026)).

**Making sense of unstructured text.** Prior work has highlighted the challenge of bridging qualitative and quantitative insights (e.g., DeVos et al. (2022); Deng et al. (2023)). Recent methodological work to this end leverages developments in LLMs (e.g., Rao et al. (2025); Movva et al. (2025); Tamkin et al. (2024); Chausson et al. (2025)), which may prove fruitful. However, we note that even more classical NLP approaches (e.g. as in Robinson (2019); Ayres et al. (2013); Bastani et al. (2019)) have shown to be effective when processing existing (natural-language) report data.

**Sequential analysis and online decisionmaking.** Statistics has studied sequential testing for decades (e.g., Wald's SPRT (Wald & Wolfowitz, 1948)). More recently, e-values (e.g., Vovk & Wang (2021)) have emerged as a popular framework for sequential approaches; however, as Dai et al. (2025a) note, nontrivial extensions for settings more complex than a single hypothesis are open technical problems. Thus, it may be fruitful to explore methods that loosen the stringency of a true sequential hypothesis test, and/or that seek to reconcile the statistical approach with the text-based methods discussed above. For instance, insights from the rich literature in online optimization and decisionmaking may offer methodological contributions to handling sequentiality.

## 6. Alternative views

This work seeks to establish *aggregated individual reporting* as a framework for post-deployment evaluation of AI systems. There are several well-founded criticisms of the approach; nevertheless, we see these concerns as important considerations for carefully designing and improving AIRs, rather than reasons that they should not be attempted at all.

We see two major categories of failure modes for AIRs: those arising from crowdsourced reporting as a data source, and those arising from organizational challenges.

**(1) Reporting is a fundamentally-flawed source of data.** By design, feedback collected by AIRs is "one-sided," and moreover, relies on usage or adoption at scale. One natural failure mode is that *individual reports would be too noisy or too biased (e.g., by reports attempting to "game" the system) as to be unusable.* This is a serious concern, especially given the existence of (online) crowd behaviors like review bombing (Payne, 2024), and the varying quality of complaints about content moderation algorithms. For instance, user feedback about moderation is often about explicit content (e.g., Movva et al. (2026)). Second, *users may not submit reports even if the mechanism technically exists—e.g., if reporting is too burdensome, or if affected populations are unaware of the option to report.* Encouraging sufficient participation is also a common challenge in settings that rely on eliciting data from the public (e.g., study recruitment and retention (Koo & Skinner, 2005)).

Handling both of these concerns is partially an empirical question (in what ways would a reporting system for evaluating AI induce different behaviors? What kinds of report affordances affect incentives?) that can only be understood by analyzing a real-world implementation. On the other hand, as mentioned in Appendix B, there are also various research communities that can contribute towards these questions.

**(2) Organizational challenges may complicate the pathway to downstream action or accountability.** One key premise of our proposal is that reports can be empowering because they can effect action, rather than languishing in an online database. However, for this to happen, there are organizational and institutional problems that must be addressed beyond technical and methodological challenges.

One clear limitation is that *model developers may not be incentive-aligned.* Even if problems with the evaluated system are identified, the system developer is not necessarily bound to address them. Though Mozilla's YouTube Regrets study was discussed earlier as a positive example of aggregation (McCrosky & Geurkink, 2021), YouTube never explicitly acknowledged the study as influencing specific choices about their recommendation algorithm, which makes it difficult to directly measure the study's impact. At the same time, a first-party administrator could evade accountability by avoiding disclosure of findings.

On the other hand, *running an AIR mechanism as a 3rd-party may be unsustainable.* Many third-party audits are foundation-funded (e.g., RegretsReporter by Mozilla), and the path to long-term financial viability is uncertain, which means that many 3rd-party auditors simply cease to exist. For example, the previously-lauded UberCheats browser extension (Marshall, 2021) no longer exists; the system that became FairFare (Calacci et al., 2026) was originally known as DriversSeat,[11] but it is unclear if prior data was ever used.

These are critical concerns about ideal patterns of implementation, especially when considering which institutions ought to play what roles. We cannot guarantee *a priori* that these failure modes can be avoided, but we hope that by emphasizing the role of organizational factors in Section 5, we can encourage intentional decisionmaking in this regard.

## 7. Calls to action

Despite these limitations, we believe that aggregated individual reporting mechanisms are a natural component of the post-deployment evaluation ecosystem. Recent events—and ongoing research—have illustrated that individuals have unique contributions to understanding the contours of AI system behavior. The constellations of individual experience are an invaluable resource not just for model development, but for understanding potentially-unintended societal consequences of already-deployed systems.

Our main call to action, therefore, is for AIRs to be studied, and to be built. *Academic researchers* should develop the methodological innovations and empirical analyses necessary for effective implementation; this is an interdisciplinary challenge that spans several subfields of computer science (and beyond), including not just AI/ML but also statistics, human-computer interaction, and law and policy. *Activists and industry practitioners* should explore paths to implementation. *Policymakers* should work in collaboration with the aforementioned stakeholders to understand what kinds of legal or policy leverage may be useful.

To translate the AIR approach from a hypothetical strategy to reality, it is essential for the area to mature; addressing the wide range of design and methodological issues outlined in this position is just the first step in coalescing a community to make progress towards this goal. But, at the same time, it is impossible to wait for all the hypothetical kinks to be smoothed: the success of AIRs is a fundamentally empirical question. We believe that it is worth trying to find out.

---

[11] www.driversseat.co/

## Acknowledgements

JD is supported in part by the National Science Foundation Graduate Research Fellowship Program under Grant No. DGE 2146752. BR is generously supported in part by NSF CIF award 2326498, NSF IIS Award 2331881, and ONR Award N00014-24-1-2531. IDR is supported by the Mozilla Foundation and MacArthur Foundation. IYC is supported by a Google Research Scholar Award and Apple Machine Learning Research Grant. The authors thank Nika Haghtalab and Paula Gradu for helpful conceptual discussions and support throughout the process.

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

# A. Additional related work

**Current approaches to post-deployment monitoring.** For understanding real-world usage of Claude, Anthropic has developed a system called Clio, which applies $k$-means clustering to a fixed batch of chat transcripts, then post-hoc labels the clusters with Claude (Tamkin et al., 2024). Unlike the incident databases described above, Clio is specific to an evaluated system (Claude)—and, notably, the goal of Clio is specifically to identify "unknown unknowns" in usage patterns. However, the collection of all chats is quite different from individually-submitted reports that highlight problematic behavior, and thus Clio captures different insights than an AIR would. For large language models in general, LMArena (formerly Chatbot Arena) has become a popular evaluation platform for ranking LLMs (Chiang et al., 2024). While LMArena does rely on real-time, crowdsourced feedback, the goal of their evaluations is primarily to compare models (via ranking), rather than conducting fine-grained evaluations that could inform future action.

**The Democracy Levels framework.** One recent work that seeks to place AI governance mechanisms in the context of "democratic" processes is the Democracy Levels framework of Ovadya et al. (2025). This framework rests on a conception of "democratic processes" as involving a remit (scope), constituent population, and an output decision; the hierarchy of "levels" corresponds to the extent to which the outcomes of this process have binding power over the AI system being governed.

The examples given in the Democracy Levels framework are primarily about *pre*-deployment rules and guidelines. However, we argue that mechanisms for aggregated individual reporting can be seen as an almost direct stand-in for their understanding of a "democratic process": the constituent population can be seen as our affected population, the remit can be thought of as our evaluated system, and the decision can be thought of the evaluation computed from the aggregated reports. (One interesting difference is that their definition of "democratic process" assumes a one-time event that can be initiated by various discrete triggers, whereas we think of our mechanism as continuous.) In fact, the mechanisms we propose can be direct drop-ins to their "levels"—for instance, in Level 0, aggregated reports have no impact; in Level 1, reports are accepted but do not necessarily affect the deployed system; in Level 2, the results of evaluation directly imply a default response action.

# B. Examples of design questions

| Design decision | Options and examples | Example research questions |
|---|---|---|
| Organizational | | |
| Mechanism administrator & relation to evaluated system | • First-party: same org (e.g. OpenAI for ChatGPT)
• Second-party: user of evaluated system (e.g., hospital system for AI scribe product)
• Third-party: External org (e.g., government or activist nonprofit) | • What arrangement is "optimal" or incentive-compatible?
• How do individuals within these organizations conceive of pathways to impact? |
| Scope of "affected population" | • System users (e.g., ChatGPT users)
• System users and those close to them (e.g. friends/family of ChatGPT users)
• System users and non-users affected by system usage (e.g., healthcare workers and patients) | • How does the inclusion of different "user roles" affect the substance of report content? |
| Individual reporting (*Reports are submitted by members of the affected population about specific experiences with the evaluated system*) | | |
| Publicizing the reporting mechanism | • Reporting option directly available at or after each system interaction (e.g., "submit report" available in UI, or sent as part of follow-up)
• Advertisements on social media about the opportunity to submit reports | • What pathway is the most effective?
• How do publicity methods affect who submits reports and why? |
| Handling reporting behavior | • Submitted reports are taken "as is" with the understanding that reporting
• Assumptions are made about reporting behavior (e.g., that heterogeneity in reporting rates is not too extreme)
• Side information (outside of submitted reports) is sought in order to characterize reporting behavior (e.g., choosing some subset of reports to "verify") | • Who is more likely to report, and why? What topics/types of problems are more likely to generate reports, and why?
• How can we model and detect disparate rates of reporting for various reasons? How can we draw inferences that account for or are robust to uncertainties in reports? |
| Report content and affordances | • About specific one-off interactions (e.g., "I said X and the model responded Y")
• About longer-run series of interactions (e.g., "over the last 2 weeks, the model has been telling my partner that they are a 'chosen one'")
• May or may not contain sufficient "state" to completely reproduce the problem (e.g., real chat transcript may or may not be available)
• May or may not contain additional contextual information about the reporter or impacted party (e.g., demographics, context on usage, location, timestamp, etc.)
• May or may not include information about believed severity, justification, or proposed solution (e.g., "this was a minor problem that could worsen"; "this was problematic due to X"; "it would have been better if Y happened instead") | • How do different affordances in report format enable reports about different types of harm?
• How can we quantitatively process different types of information, including rich unstructured data like free text, and incorporate them into a quantitative (possibly-statistical) aggregation? |
| Incentivizing and encouraging reports | • No additional incentive for reporters
• Financial compensation for reports that turn out to reflect what is later deemed to be a 'true' problem
• Report affordances are designed to shape reporting behavior (e.g., intermediate report summaries are made visible to the public, so that reporters may be interested in contributing to the conversation about currently-leading concerns) | • What motivates potential reporters to actually submit reports?
• Are there concerns about misaligned incentives that might affect the 'quality' of reports?
• Are there feedback loops that might arise? |

*Figure 3. Organizational and interaction-focused questions.*

| Design decision | Options and examples | Example research questions |
|---|---|---|
| Online aggregation (*Reports are aggregated and interpreted over time; the goal is describing system behavior in a fine-grained manner*) | | |
| Evaluation trigger (what evaluation seeks to identify, and what would trigger downstream action) | • Set up with specific type of harm in mind; try to identify the specific harm as soon as possible if it does arise
• Set up with a more unstructured approach; try to identify important trends that emerge over time, and allow | • What methods can successfully achieve these goals in an online or quasi-online manner? How do they adapt to or handle changes over time?
• What methods can adapt to richer "hypotheses"?
• How can we balance batching data with handling true sequentiality? |
| Type of evidence as an output of aggregation | • A rigorous statistical guarantee is required (e.g., "with probability 1-α, the findings of the aggregation are significant…")
• More ad-hoc interpretation is sufficient (e.g., looking at the output of a clustering algorithm) | • What are the relative strengths and weaknesses of various types of evidence for the purpose of motivating downstream action?
• What are the tradeoffs involved in pursuing more vs. less "rigorous" outcomes? What methods strike a balance between data efficiency vs. validity of conclusion? |
| Mechanism for action (*Some evaluation results may trigger action; the mechanism administrator is responsible for doing so.*) | | |
| Types of action that can be taken once evaluation trigger is reached | • First-party administrators: Rollback to prior model version (e.g., if new problems suddenly arise after new deployment)
• Second-party administrators: Revisit usage policy (e.g., internal guidelines for when a tool should be used or not) and/or provide feedback to owner of evaluated system
• Third-party administrators: Build public pressure (e.g., via media) and/or action towards accountability (e.g., legal case against first-party)
• All administrators: may use the evaluation trigger as a starting point for further investigation (e.g., additional data collection) | • What kinds of evidence can compel action from an external third party?
• How can closed systems be pressured to admit a reporting system? |
| Ongoing contact with affected population | • Public notice of problem identified & changes made
• Specific outreach to reporters who noticed the problem | • To what extent does, and should, the reporting mechanism encourage long-term engagement from reporters? |

*Figure 4. Methodology-focused questions.*

