# OpenReview forum: "Position: Mechanisms for Aggregated Individual Reporting Should be Established for Post-Deployment Evaluation"
_ICML.cc/2026/Position_Paper_Track — ICML 2026 Position Paper Track regular_

### Official Review · Reviewer_DXHU · 2026-03-08

**Significance:** 2
**Argument Clarity:** 3
**Rating:** 4
**Confidence:** 3

**Questions:**

See the weakness section

**Alternative Views Section:**

No

**Compliance With Llm Reviewing Policy A Conservative:**

Affirmed.

**Discussion Potential:**

2

**Final Justification:**

My main concerns are fully addressed.

**Paper Summary:**

This paper advocates aggregated individual reporting (AIR) for deployed AI systems such as LLM-based products, drawing on lessons from existing AIR practices in domains such as clinical reporting, loan allocation, and consumer protection. The paper reviews current AI reporting mechanisms—categorized as incident datasets, flaw disclosure, and bug bounty programs—and proposes three criteria for AIR systems: individual reporting, aggregation for evaluation, and evaluation-conditional action.

**Position:**

Yes

**Position In Title:**

Yes

**Related Work:**

3

**Strengths And Weaknesses:**

Strengths
1. Clear writing: The story-based introduction and well-organized section structure make the paper easy to follow and engaging to read.

Weakness
1. Missing “Alternative Views” section: The paper lacks a clearly identified Alternative Views section, which is mandatory.
2. Unclear relevance to ICML: The paper does not clearly justify why it is submitted to a machine learning conference. Much of the discussion reads like a policy proposal for governmental or regulatory action. It would help to clarify what ML researchers are expected to learn or contribute from this work.
3. Weak empirical example: While the paper discusses many examples from domains such as healthcare and loan allocation, the AI-specific example is limited to a single case of sycophantic behavior in ChatGPT. As a result, the discussion feels dominated by loosely related literature reviews rather than concrete AI system analyses.
4. Practicality concerns: Simply asking users to provide more detailed reports is unlikely to scale, particularly when reporting is voluntary. Effective systems typically reduce user burden. Additionally, the paper does not address who would maintain such a system or what incentives would exist for maintaining it.
5. Unclear novelty of aggregation: Aggregation appears to function primarily as preprocessing of reports. It is unclear why the proposed criteria would exclude or substantially differ from existing mechanisms such as bug bounty programs or red-teaming. For example, it is not obvious why similar aggregation could not be achieved via simple tagging (e.g., hashtags) followed by post-hoc analysis.
6. Normative assumptions in aggregation: The paper implicitly assumes that aggregate counts reflect the plausibility or importance of issues, which corresponds to a utilitarian perspective. However, this may fail in cases that are rare but highly consequential (e.g., safety-critical failures), where egalitarian or risk-sensitive principles may be more appropriate. In general, designing an aggregation rule that satisfies all fairness or democratic criteria is impossible (e.g., Arrow’s impossibility theorem). Thus, the critique of quasi-democratic aggregation in prior work may also apply to the proposed framework.

Minor Comments
1. Figure placement: Figure 1 appears far from its first reference in the text.
2. Definition of affected population: It is unclear whether ChatGPT users should be treated as a specific “affected population,” since in practice this group may include a very broad portion of the public.
3. Assumption about social media incentives: The discussion assumes that users post on Twitter/X primarily for virality, which may not reflect all users’ motivations.

**Support:**

2

---

> ### Author Rebuttal · Authors · 2026-03-31
>
> Thanks for the detailed engagement! We'll address your concerns as follows.
>
> **W1: Alternative views.** Section 6.1 contains the expected content for the alternative views section ("describes and addresses one or more credible (not strawmen) positions that are opposed to the paper’s position", per the CFP).
>
> **W2: ICML relevance.** AI evaluation is directly relevant to an ICML audience; we submitted our work to the position paper track because it expresses a position about machine learning research and practice. For example:
> * Sections 3 and 4 engage directly with active conversations in the machine learning research community.
> * Our CTA (Section 6.2) explicitly includes a role for academics.
> * In Appendix C and Figures 3-4, we give 20 concrete research questions and highlight 4 distinct research communities that have relevant expertise to contribute.
>
> Moreover, the CFP lists "Regulation of ML technology (licensing, evaluation, disclosures, post-deployment monitoring, etc.)" as explicitly in scope.
> For example, prior papers published at the position paper track at just last year's ICML include:
> * Longpre et al., "Position: In-House Evaluation Is Not Enough" (https://openreview.net/forum?id=ACzL62Jp4E), which gives a similar proposal to ours, but is more explicitly focused on policy context and has less researcher-oriented content, and was presented as a spotlight.
> * Appel, "Position: Generative AI Regulation Can Learn from Social Media Regulation" (https://openreview.net/forum?id=fRk0nKLKrJ)
> * Ovadya et al, "Position: Democratic AI is Possible. The Democracy Levels Framework Shows How It Might Work." (https://openreview.net/forum?id=yYJo8czj4f)
> * Mushkani et al., "Position: The Right to AI" (https://openreview.net/forum?id=IxCvgUme5S)
>
> **W3: empirical examples.** Our introductory example is AI sycophancy. However, we think AIRs can and should work for more than just chatbots; Section 5 describes two examples for AIRs that involve AI systems, one in a loan allocation context and one in a health context.
> More broadly, as we note in L101-103, AIRs are not currently in mainstream use for AI systems, which means there do not exist any 'real-world' examples that are about both AI systems and actual reporting mechanisms (like VAERS). We hope our paper can help to change that.
>
> **W4a: Unlikely to scale... when reporting is voluntary.** We agree that this is a challenge and discuss it explicitly in L399-423. Additionally, the examples discussed in S3 suggest that members of the public _are_ often willing to share their experiences, both informally (as in the case of social media) and in purpose-built reporting systems (as in the case of VAERS/CFPB).
> In Appendix C, we cite some examples of work that studies what motivates participants in online platforms (L742-748) and give examples of research questions related to voluntary reporting (Fig 3, column 3).
>
> **W4b: Does not address who would maintain such a system.** Different AIRs can have different administrators (see L70-L78c2 and Fig. 1, col 2). We think this is an important aspect to get right, and explicitly discuss various considerations for such a decision, including incentive-related limitations (L326c1 - L336c2 and L424c1 - L403c2). Overall, while the most enduring organizations for maintaining reporting systems seem to be governmental, it is plausible that NGOs can be effective as well (see our response to BQVa).
>
> **W5: Novelty of aggregation method.** It is possible that in some cases, existing methods are sufficient for aggregation; however, work is still needed to apply them to AIRs. Additionally, open questions remain. See, Figures 3-4, right column; new methods may need to take reporting behavior into account, for example. We discuss the differences from red-teaming and bug bounties on L110-121.
>
> **W6: Normative assumptions**. As mentioned at the end of S2, we do not think AIRs can or should cover all relevant problems, and other eval approaches should handle (e.g.) safety-critical failures. From our point of view, in the same way that the existence of Arrow's theorem does not mean that we stop trying to have elections, the existence of limitations in a particular aggregation approach does not mean we stop trying to conduct the evaluation.
>
> **Minor comment 1: figure.** We will use the extra page in the camera-ready to adjust formatting.
>
> **Minor comment 2: definition of affected population.** We see no issue with defining the "affected population" as a large portion of the public; anyone who interacts with ChatGPT can reasonably form an opinion about their experiences with it. For any given harm, it is likely that the subset of the "affected population" that actually experiences that harm is much smaller.
>
> **Minor comment 3: Social media incentives.** To clarify, we do not assume that users are motivated by virality. The discussion in L191-208 is about how sycophancy-related content was friendly to the _algorithm_ and thus easily went viral.

---

> > ### Author Rebuttal · Reviewer_DXHU · 2026-04-03
> >
> > The authors have addressed my concerns well. I have updated my score to weak accept.

---

### Official Review · Reviewer_P5db · 2026-03-13

**Significance:** 2
**Argument Clarity:** 1
**Rating:** 3
**Confidence:** 4

**Questions:**

See above

**Alternative Views Section:**

Yes

**Compliance With Llm Reviewing Policy A Conservative:**

Affirmed.

**Discussion Potential:**

1

**Final Justification:**

Some of my concerns were resolved, and the idea seems interesting. However, still some unanswered questions remain and will be helpful to address them for the clarity of the reader.

**Paper Summary:**

The paper proposes a new framework for evaluating AI systems after deployment with real user reports. The paper highlights the the issue in traditional evaluations with benchmarks since they cannot capture sevral real-world failures. The framework has three key elements: individual reporting, aggregation, and action triggered by evaluation results and is inspired by reporting systems in other domains such as vaccine safety monitoring etc. The main claim is that aggregated reports can reveal unknown failure modes, safety risks, or unfair impacts on several user groups which might not be easily detected through reporting in various social medial platforms.

**Position:**

Yes

**Position In Title:**

Yes

**Related Work:**

2

**Strengths And Weaknesses:**

The AIR framework is easy to understand and also easy to deploy in the UI of the AI systems like ChatGPT, where the user can report the issue. This can be a simple addition to the sytems which can help in faster understanding of the challenges or issues if faced by multiple users. Although the idea is interesting, the significance and criticality of the problem is not well understood, so the authors should cite use cases which have been delayed and could have been expedited if this system was there. Secondly, the incentive is not very clear unless the problem is very critical. Also, under sensitive scenarios users might not be incentivized to provide sensitive data.
For example, if the AI system is biased towards certain groups, the other groups might not want to reveal their sensitive information to mitigate such bias. Misreporting issues can be a significant problem, unless the users are incentivized to do and would require methods to ensure correct data is being propagated which is extremely challenging.
Additionally, the paper is extremely conceptual and high level and needs concreteness in providing some key metrics to evaluate the information is correct or some more concretizations through examples, on how this reporting can improve the performance. Additionally as these AI systems are being used by millions of users across the world, reporting issues require significant compute, so a clear discussion of infrastructure, changes in architecture, compute frameworks is needed which is missing.

**Support:**

2

---

> ### Author Rebuttal · Authors · 2026-03-31
>
> Thanks for reading and engaging with our work! We'll do our best to address your concerns below, though it would help for us to understand which parts of the argument you felt were unclear.
>
> **The significance and criticality of the problem is not well understood.** Section 3 provides the practical motivation with concrete examples throughout: without individual reporting, it is impossible to identify unknown unknowns, and without aggregation, it is difficult to pursue accountability. Section 4 provides the normative motivation. Could you clarify why the problem did not seem significant or critical?
>
> **the authors should cite use cases which have been delayed and could have been expedited if this system was there.** We discuss many examples of potential AIR use cases, including:
> * AI sycophancy, and why that was a special case (L192-208, column 2);
> * Loan allocation (Fig 1 and Section 5)
> * Medical scribe product (Fig 1 and Section 5)
>
> We also discuss examples of non-AI systems that deal with crowdsourced reports, including:
> * VAERS (L222-246, column 2);
> * CFPB (L251-271, column 1);
> * RegretsReporter and Fairfare (L286-321, column 1)
>
> **Secondly, the incentive is not very clear unless the problem is very critical...would require methods to ensure correct data.... which is challenging.** We agree incentives are important; we discuss this in the paper explicitly throughout Section 5 (L351-362), Section 6.1 (L399-422), and Appendix C (L742-748 and Figure 3). We think they are a solvable problem, especially given active research communities in mechanism design, statistics, etc. These are questions best handled in separate technical work, as high-quality analysis of incentive structures requires careful modeling of agent behavior, mechanism priorities, and so on that is beyond the scope of a position paper.
>
> **the paper is conceptual and high level.** Our goal in writing a position paper is exactly to give the conceptual and high-level arguments that a technical main-track paper would not. We provide concrete examples and connect to relevant academic conversations throughout. Which parts of the paper do you wish had more examples?
>
> **reporting issues require significant compute.** We disagree. AIRs don't require training or inference of large AI models, and storage is cheap. Even at the largest scale, we would not require different resources from a typical software company; compute would not require new frameworks, architecture, or infrastructure.

---

> > ### Author Rebuttal · Reviewer_P5db · 2026-04-04
> >
> > Thanks for clarifying the doubts. It helped in clearing some of my concerns.
> > But still few of my concerns remains
> > 1. The framework assumes good-faith users, however a majority can be a hack to just jailbreak the system. So concrete discussion and mitigation strategies needs to be discussed.
> > 2. Second lies a big assumption - Selection bias since only a very few users might report? They might just use some other AI system, if its not a global issue unless there is a proper incentive. Hence, i asked previously to provide some clear example of incentive in the specific context.
> > 3. The paper has too many analogies and is over-reliant on analogies (e.g., vaccine systems). It will be helpful to provide clear examples which are very specific to current LLMs, AI Agents? I am not able to properly relate to the above analogies
> >
> > For example- Lets say a Math agent, a coding agent, a tool-use agent, Web agent ? These are the AI agents used most
> >
> > Can you give some 1-2 examples for the above where individual reporting can help? It will extremely help my understanding.? Even if its in the paper and I have missed, please point me and I will re-read it.
> >
> > P.S - I am happy to increase the score, if i fully understand the significance of this in the context of the current AI agents and use-cases.

---

### Official Review · Reviewer_BQVa · 2026-03-16

**Significance:** 4
**Argument Clarity:** 3
**Rating:** 5
**Confidence:** 4

**Questions:**

What technical mechanisms exist to limit or prevent gaming?

What incentive structures could be set up to allow third-parties to set up AIR systems that are sustainable in the long term? What, if anything, has worked in the past?

**Alternative Views Section:**

No

**Compliance With Llm Reviewing Policy A Conservative:**

Affirmed.

**Discussion Potential:**

3

**Final Justification:**

Much about the other reviews and the authors’ rebuttal reinforced the judgments expressed in my original review. The paper should be accepted.

**Paper Summary:**

The paper argues for the establishment of a greater number of systems for aggregated individual reporting (AIR) from users of online systems such as AI models.

**Position:**

Yes

**Position In Title:**

Yes

**Related Work:**

4

**Strengths And Weaknesses:**

Overall, the authors provide a highly effective discussion of a useful proposal. I was skeptical when I started reading, but the depth of discussion and the utility of the basic proposal won me over. At least in my experience, aggregated individual reporting is overlooked, but very important. Specifically, the paper provides a clear description of the general concept, a very useful taxonomy of possible AIR system designs, and very useful accounts of existing systems that are similar to what the authors discuss.

A few comments on potential improvements:

Many feedback mechanisms, including the one proposed here, could be *gamed*. Some discussion of this issue is provided in Section 6.1, but this discussion is relatively brief, particularly given the empirical experience that already appears to exist with reporting systems. A more extended discussion would be useful, particularly about when such “gaming” can be expected and when it is likely to minimal.

*Incentives* are another issue which is not discussed at length (though, again, a very brief discussion is provided in Section 6.1). Who, besides the developer of a given system, has incentives to support and manage such reporting? In some cases (as the authors describe), a government can support such reporting. However, the paper seems to imagine cases in which third-party organizations might support reporting. In that case, I’m unclear how such support could be sustained.

Another issue that would be useful to discuss is *fragmentation*. If many third-party organizations support different reporting systems, then users might not know which system to use and data would be spread across multiple organizations. This is issue is probably less serious than gaming and incentives, but it is still worth a mention.

A minor issue: Near the end of Section 1, the authors are inconsistent in how they reference government actions. They refer to “the EU AI Act (European Parliament, 2024), the UN General Assembly’s first AI resolution (Assembly, 2024), and Biden’s now-repealed executive order (Biden, 2023))…” I would, at least, say “U.S. President Biden”. Even better might be “…and a now-repealed executive order by the U.S. President (Biden, 2023)...” Also, see style guides about citing executive orders. They are usually cited with something like: (Executive Order No. 13,231, 2002) rather than naming a specific president.

Finally, the "Discussion" section is very similar to the mandatory "Alternative Views" section, and could be retitled to this.

**Support:**

4

---

> ### Author Rebuttal · Authors · 2026-03-31
>
> Thanks for reading and the feedback -- we'll make suggested revisions! We agree incentives and gaming are probably the biggest fundamental question marks around how to do this well, and we hope our manuscript can serve as a starting point for a productive conversation about these details.
>
> **Q1: gaming.** At the platform/product level, review bombing, brigading, and other types of coordinated adversarial activity is a well-known problem in current online platforms, and we're hopeful that existing techniques can be relevant here. The reporting interface itself can be gated; VAERS, for instance, mostly avoids brigading despite being about a very sensitive topic because there are some barriers regarding who can submit problems and how.
>
> That said, we're also excited about the technical front here; there's a clear set of mechanism design problems, e.g. how to strongly disincentivize 'lying' (or strongly incentivize truthfulness) in a report. One way to do this might be, e.g., conducting random verification checks for some subset of reports, or by rewarding high-quality/high-effort reports (for some measure of quality/effort). We're also interested in the inference-meets-incentives literature from statistics for methods that are not just robust to random noise but handles non-'iid-ness' arising from correlated or quasi-adversarial behavior.
>
> **Q2: organizational incentives.** The most enduring or effective analogous systems, to the best of our knowledge, seem to be run at the government level --- vaccine, pharmaceutical, product safety, aviation, cybersecurity, environment etc. (see also Gailmard et al. for more detailed discussion about many of these existing systems). At a minimum, a non-governmental organization would need some guarantee of long-run financial viability (e.g., via an endowed fund, rather than relying on repeated cycles of philanthropy fundraising or crowdfunding), while also maintaining independence (i.e., without accepting funds that appear conditional).
>
> We agree this is challenging in general to achieve for third-parties, but there do exist strong mission-driven nonprofits that satisfy those criteria -- for example, some legal aid organizations (e.g. ACLU, SPLC) and media outlets (e.g., ProPublica, Guardian, The Markup) have clear long-run organizational mandates and quasi-endowment funding -- that give us hope that such organizational structures aren't impossible for an organization with an AI auditing/eval goal.

---

> > ### Author Rebuttal · Reviewer_BQVa · 2026-04-05
> >
> > My (few) concerns are acknowledged and addressed by the authors with an agreement to revise the paper accordingly.

---

### Official Review · Reviewer_WGcv · 2026-03-23

**Significance:** 3
**Argument Clarity:** 2
**Rating:** 5
**Confidence:** 4

**Questions:**

How should AIR’s sequential hypothesis test be reset or adjusted  when the model I rolled back or updated ?

We know that different subpopulations report issues at different rates, what specific statistical methods would you recommend to ensure lack of reports from specific group isn’t misrepresented as a lack of harm ?

What safeguards are necessary to prevent review bombing or coordinated adversarial attacks meant to trigger a false positive model rollback ?

**Alternative Views Section:**

No

**Compliance With Llm Reviewing Policy A Conservative:**

Affirmed.

**Discussion Potential:**

3

**Paper Summary:**

This position paper argues about the establishment of aggregated individual reporting framework as one of the critical aspects of post-deployment evaluation schemes. In attempt to move beyond static benchmarking. The proposition is that harms of AI can be turned into actionable statistical evidences.

**Position:**

Yes

**Position In Title:**

Yes

**Related Work:**

3

**Strengths And Weaknesses:**

Strengths of the paper:

1. It is very timely and effective. The authors use the GPT-4o sycophancy update as a motivating case study, which is great where informal social media feedback led to a model rollback.
2. The authors also bridge the gap between concepts of HCI , statistics and political science.
3. Lastly, they provide a clear breakdown of the organizational roles and their tradeoff regarding data access and alignment with incentivization.

However, the paper has some weaknesses such as:

1. First party administrator such as the model developers may not fully solve or disclose the negative findings.
2. As highlighted in the UberCheats and Fairfare discussion, third party reporting systems often struggle with long-term financial viability once the funding dries up - which is critical for continuous monitoring and maintenance.

**Support:**

2

---

> ### Author Rebuttal · Authors · 2026-03-31
>
> Thanks for reading and engaging with our work! We agree there are lots of questions around how to implement AIRs effectively, and we hope our manuscript can serve as a starting point for a productive conversation about these details.
>
> **Q1: adjusting tests with rollbacks/updates.** This would likely depend on the specific algorithm/test used by the AIR and design choices made by the administrator. From a purely statistical point of view, if a betting-style/e-value anytime-valid test is used, there should be no issue with "resetting" hypotheses to test against a historical baseline as long as the test statistic itself uses only fresh data.
>
> **Q2: (under)reporting.** We think handling varying reporting rates is one of the key challenges of moving AIRs forward. In the Dai et al. case study from ICML25, they make an assumption that reporting behavior does not vary too drastically by subpopulation, which allows them to make a claim that says the true degree of relative harm experienced by a particular group is inversely proportional to the degree to which they tend to underreport. That is, if some subgroup appears to experience harm at level X, but they report half as often as average, then their actual level of harm is 2X.
>
> However, if a subpopulation never reports, then no mechanism can accurately determine whether that subpopulation experiences problems. More sophisticated technical approaches might involve, e.g., careful mechanism design to incentivize submission of feedback from subpopulations that the mechanism currently lacks information about, or active sampling of data from those subpopulations; however, for these approaches, one would need to be careful about whether such designs might also induce unexpected changes in reporting behavior among the population more broadly.
>
> **Q3: safeguards against coordinated attacks.** Luckily, review bombing, brigading, and other types of coordinated adversarial activity is a well-known problem in current online platforms, and we're hopeful that existing techniques can be relevant here. One can also imagine mechanisms designed to be incentive-compatible so that 'lying' in a report would be strongly disincentivized, e.g. random verification checks for some subset of reports. One other thought is that the reporting interface itself can be gated; VAERS, for instance, mostly avoids brigading despite being about a very sensitive topic because there are some barriers regarding who can submit problems and how.

---

### Decision · Program_Chairs · 2026-04-30

**Decision:**

Accept (regular)

**Comment:**

The decision is to accept the paper.

The paper recommends formalizing user experience reporting for AI systems in aggregated individual reporting (AIR) systems. The authors consider may aspects of how such systems could be implemented, and give examples from other domains (e.g., vaccine adverse events) where such reporting systems have been used effectively. The authors are also open about what aspects for AIRs still require additional research.

While there was some disagreement about whether the proposal was sufficiently concrete, reviewers generally agreed that AIRs address a critical problem, and represent a useful solution class. The depth of discussion in the paper and connection to real world experience was noted as a particular strong point. Reviewers raised specific concerns about mechanism design, potential organizational challenges, and some other issues. The authors addressed these adequately rebuttal, but the consistent experience across reviewers suggests that some of these issues could be addressed in greater depth in the camera ready.